# Social Capital as an Inclusion Tool from a Solidarity Finance Angle

**Juanita Salinas** [1,*] **and Susana Sastre-Merino** [2,*]

1    Student of the Doctoral Program in Planning of Rural Development and Sustainable Management Projects, Universidad Politécnica de Madrid, 28040 Madrid, Spain
2    Institute of Education Sciences, Universidad Politécnica de Madrid, 28040 Madrid, Spain
*    Correspondence: jv.salinas@alumnos.upm.es (J.S.); susana.sastre@upm.es (S.S.-M.); Tel.: +593-999-542-187 (J.S.)

**Abstract:** Within rural environments, the construction of financial ecosystems that both stimulate local development and contribute to poverty reduction requires an increase in associative community activity. Such activity serves as a fundamental means of organizing territorial production systems, reinforcing capacities, and strengthening the negotiating position of the population being offered financial services. Solidarity finance is important because it recognizes that collective action and criteria such as social efficiency, local capacities, cooperation, associativity, the social fabric, self-management, and resource recirculation are integral aspects of financial evaluation. Therefore, this research proposes a methodology to reinforce the financial service delivery of solidarity finance institutions through the evaluation of social capital in rural production organizations. Social capital is regarded as a resource of the organization's constituents that can facilitate financial inclusion and generate value for rural populations.

**Keywords:** social capital; financial inclusion; solidarity finances





## 1. Introduction

In the present context of accelerated and complex transformation, economic and social exclusion are global problems. The world community must re-evaluate its economic principles and include variables that prioritize environmental wellness and sustainability from the perspective of human activity [1–5].

Different perspectives and emphases on development have identified the imperative of rebuilding human dignity and creating sustainable and inclusive environments of social cohabitation [6–8]. The neo-institutional view emphasizes the unavoidable responsibility of the state to plan and manage redistributive social investments that generate development, satisfy basic needs, and guarantee rights. In contrast, perspectives that focus on community and local development highlight the endogenous nature of the process, the transformative and innovative role of the social aspect, and community capacities in the territory [9–13].

In this process, financial inclusion is a useful tool for driving practical strategies of convergent development. The improvement of living conditions requires the expansion of affordable and convenient access to a range of products and regulated financial services as well as increasing the use capacity of different population segments by considering the possibility of reinforcing the intertemporal management of resources, altering the consumer profile, managing risks, releasing investment resources, and, in short, providing means, instruments, and financial strategies to the affected population [14–16].

However, the ambiguity in the socio-economic results of intensifying the scope and depth of financial services and the danger of causing greater vulnerability and over-indebtedness for the population, question the convenience of promoting financial inclusion actions from market effectiveness approaches [17–21].

Financial inclusion efforts have advanced with microcredit and microfinance proposals from the end of the 20th century toward the need to build inclusive financial ecosystems as part of broad processes of sustainable development, the formulation of public policies and implementation of meso and micro actions for capacity development, and value generation for the population [22–24].

In rural areas, community, associative, and/or cooperative solidarity finance institutions are projected as an alternative for financial inclusion to address situations of greater exposure to productive or market risk, high population dispersion, or the existence of poor environments and low-income [25].

Solidarity finance institutions operate under the principles of equity and solidarity. Their democratic and self-managed government structures seek to use the symbolic capacity of the currency and the potential of financial services to value the accumulated work in the territories, dynamize transactions, encourage investment, and facilitate access to economic means required to meet the needs of their members and promote local development [26].

Therefore, it is essential to locate this research in the operation of solidarity finances (savings and credit cooperatives, community banks, solidarity banks, among others), since their action logic raises a political proposal of financial services that reinvents the conception of wealth within the framework of solidarity principles and the fight against domination and inequalities [27].

However, solidarity finance institutions face difficulties in incorporating community principles of solidarity action. Their primary methodologies continue to focus on evaluating the effectiveness of service delivery and the definition of risk profiles according to individuals' economic capacities, detached from and independent of the operating context [28]. This procedure is a proper feature of the institutionalized individualism denounced by Beck and Beck-Gernsheim [29], which represents a problem for solidary finances because it hinders the comprehension of the social phenomena, disincorporates the economic from the social, and does not know about community aspects and associative work practices that sustain the satisfaction of needs and the social reproduction of the rural population [30].

For small rural production, risk is associated with the probability and potential loss of well-being implied by the occurrence of an unwanted event [31]; whereas for financial institutions, the risk in the provision of services refers to the disturbance on the normal return of the investments made [32,33].

Authors such as Guiso et al. [34] and Bebbington [35], Gutiérrez [36] among others, have highlighted the importance of considering the social capital endowment of people who are part of an organization or group. Confidence and other components that are integrated in the concept of social capital contribute to the creation of financial inclusion processes that benefit residents and provide a sustainable population for financial institutions.

Therefore, solidarity finance institutions could reinforce their solidarity financial identities and greatly encourage the generation of inclusive financial ecosystems in their rural environments by developing valuation methods that account for connections and networks such as social capital and relationships in the organized population [35,37–40].

For people in rural production groups or organizations, social capital represents part of their immaterial assets, which creates a capacity for collective action that enables the responsible use of financial services, provides access to resources, and promotes the identification of new ways to satisfy needs [41].

For solidary finance institutions, valuing the social capital endowment of their members of productive organizations is meant to favor the effectiveness of the intermediation activity, incorporating principles of reciprocity and cooperation of common activities that also facilitates the availability and compliance of the contracts, reduces the cost of transactions, strengths the economic analysis, and provides enough information to face problems of adverse selection and moral risk associated with financial operations [42].

In financial analysis, the concepts need a quantitative expression that allows its incorporation as a category and variable to be considered in the evaluation processes [43].

This is why it is necessary to have a process that identifies and values the social capital endowment of productive organizations.

Social capital is a complex concept and its measurement requires finding a theoretical framework that defines the construct and operates the indicator selection in different fields, components, and variables of its symbolic expression, both structural and institutional.

Ostrom and Ahn [44] identified two general methods for measuring social capital: methods centered on minimalistic interactions among individuals and expansionist approaches that evaluate the existence of stable and reciprocal relationships in clearly delimitated collectives. Minimalistic measures include the purposes and tools developed by Portes [45]; Burt [46]; Onyx [47]; Lin, Fu, and Hsung [48]; and Van Der Gaag and Snijders [49]. Among the expansionist approaches are the works of Putnam [50] and Fukuyama [51].

Far from considering this focus as antagonistic postures in relation to the social capital definition, their approaches are complementary for identifying the multidimensional and complexity of the concept [52].

Within the rural context, small producers that organize their work collectively play an important role in the creation of social capital in the form of relationships, obligations, and reciprocal expectations. These functions are essential because they allow for the satisfaction of needs, reduced management and administration costs, increased production scale, and application of social and political pressure to access markets, public property, resources, and specialized inputs as well as to launch production and social projects that would be impossible for an individual [53,54].

IFAD indicates that small family producers who own very little land and operate on a small scale of production represent approximately 85% of all agricultural operations in the world [55].

Under different structures or grouping types, rural production organizations have established particular methods of resource administration and government based on internal rules that define the participation level, democracy, management transparency, work division, and rights and responsibilities of their partners [56]. As a setting for collective action concentrated among group members working for a common benefit, organizational solidarity depends on the quality and wealth of social relationships within the group and on the group's capacity to establish external connections that beneficially position its members in the institutional context [57].

Cooperation and/or conflict, tension, and rivalry situations are all likely in organizational relationships, and they directly affect the collective action capacity [58–61] Therefore, identifying the social capital endowment of an organization requires the incorporation of both positive and negative relationship experiences from the actors' perspectives [45,62].

Positive social capital endowment provides conditions that are conducive to an inclusive financial process and reinforces the population's capacity to engage in social learning [63], improve risk management, supply conventional contracts, function within collateral relationships, and work as a trustworthy source of information This perspective highlights the importance of evaluating the social capital of a rural production organization and including it in the financial service delivery methodology as a resource that supports the activity of small rural producers.

The concept established by Ostrom and Ahn [44] and Woolcock and Narayan [64] defines social capital as the rules and social networks that facilitate collective action by promoting decision making, establishing goals, acting collaboratively to fulfill common objectives, defining the components and variables that form the concept, recording the value and quality of the relationships that constitute the social fabric of the group or organization, and identifying their social capital endowment.

Ostrom [65] indicates that building collective action capacity not only requires that individuals cooperatively organize around common objectives, but also demands the deployment of certain attributes of mutual recognition, confidence, reciprocity, and shared identity, that is, the cognitive components of social capital, which are in line with the

group's established objectives. Ostrom and Ahn [44] refer to social capital as the field of relations where power is exercised, the field of politics, and the space for the exercise of the system of explicit and tacit rules that regulate the organizational possibilities of adaptation and change.

On the other hand, Woolcok and Narayan [64] proposed that collective action is enabled by relational components at different levels: close relationships, union ties, and horizontal interactions among the collective members or egocentric and social-centric networks are identified as bonding social capital. Diffuse links established between paired actors of different collectives are designated as bridging social capital, and relationships based on synergy and integrity that develop within the institutional framework are defined as linking social capital.

Thus, the strengths of the social capital of a rural production organization are based on complex interactions among social components of the internal network structure as well as the connections and links with external actors and the synergic relationship with the institution and the state.

With the above considerations, this work had the following objectives: (1) to design a methodology for the valuation of the social capital of rural production organizations that expands risk evaluation and contributes to solidarity finance efforts to create inclusive financial ecosystems; and (2) to apply the proposed valuation method of social capital endowment to a community organization: The Junta Administradora de Agua Potable y Saneamiento Ambiental Proyecto Nero of Ecuador.

## 2. Methodology

First, a method was designed and validated for the identification and evaluation of the social capital endowment of rural production organizations with the aim of expanding risk evaluation criteria and contributing to the effectiveness of solidarity finance, thereby improving the living conditions of the rural population. Subsequently, to illustrate the implementation of the proposed method, it was applied to a case study: The Junta Administradora de Agua Potable y Saneamiento Ambiental Proyecto Nero of Ecuador.

The proposed method for evaluating the social capital endowment of a rural production organization was incorporated into the risk evaluation methodology used by solidarity finance institutions.

The analysis unit was a rural production organization that operates in a specific context. This research was characterized by its analytical, nonstatistical generalization nature, which has been applied both historically and recently for evaluating aspects of complex relational aspects of organizations [66].

The methodological treatment is inductive, exploratory, and descriptive; it also addresses social capital evaluation with a synergic, multidimensional, and interdisciplinary focus. It incorporates a cognitive dimension and structural and institutional variables of social capital from a socioeconomic perspective [67], and identifies the relational capacity of the organization to undertake collective action, make decisions, establish goals, and obtain common benefits while accounting for complex interactions among the social components of the internal structure and the external conditions of the sociostructural context. On balance, these conditions determine the existence of additional advantages for members who are part of an organization compared to their individual actions [45,46,68–72].

The proposal was located within the conceptual framework of the commons developed by Ostrom [65,73]. The author identified general principles of government that support the effective management of different forms of the self-organization of collective action. She affirms that the institutionalization of practical rules that encourage cooperation and respect for what is established requires a high degree of social capital on the part of the stakeholders.

The content and scope of the concepts incorporated in the proposal allow us to identify the social capital of rural productive organizations that act under different conditions around the world. Its application to particular cases requires a prior critical analysis of the

historical-social reality of the group's context to adapt the instruments and adjust the set of observable indicators to the particular case study [74].

It is necessary to consider that although the dimensions, components, and variables used by the proposal to assess the social capital of rural productive organizations are generalizable to different contexts of action, the selection of indicators depends on their pertinence and ability to explain the category under the particular circumstances of the organization's performance. Therefore, the indicators are not generalizable, and it is not possible to have a pre-set battery or specify the number to use, so they should be identified from the critical analysis of the specific historical-social context.

The identification of the analysis categories resulted from a reflective co-design process that included dialogue with members and leaders of the organizations to select and define the indicators to be used. In addition, experts from the field of research and academia contributed to improving the categorization of the dimensions, components, and variables considered by the proposal. Finally, the proposal received the contribution of solidarity finance institutions that made their service delivery methodology transparent and thus made it possible to locate the possibility of incorporating the evaluation of social capital in their service procedures. Joint workshops were held between the researchers and the other two actors. In these workshops, the results of the model created from the bibliographic review were presented, and from this, the relevant elements were agreed upon. Regarding community organizations, two workshops were held with each of the following organizations: La Unión, Nuevo Mundo, Serrag-Ludo, Asosercabo, Cufe and Ñucanchi Huasi, and the Junta Administradora de Agua Potable y Saneamiento Ambiental Proyecto Nero of Ecuador during the months of March, April, and May 2017. In relation to solidarity finance institutions, the methodology for providing services was analyzed through five work sessions with researchers and financial services personnel from the Jardín Azuayo Savings and Credit Cooperative.

The analysis of cognitive, structural, and institutional dimensions of social capital involves different relational components and variables of the organization in the territorial and operating contexts, the organization profile, and the relationships among members. Table 1 lists the different measurement instruments, the sources of information, and the research tools used to identify significant aspects of the social capital of rural production organizations.

**Table 1.** Research techniques applied for the social capital measurement of rural production organizations.

| Field | Measurement Tool | Information Source | | Research Tools | |
|---|---|---|---|---|---|
| Territorial and operating context | Record of the organization's territorial and operating context | a. | Integral Land Management Plan and Locality Integral Diagnosis. | a. | Documented evidence systematization |
| | | b. | Documented registration of work formalization among the organization and the local or central government. | b. | Joint interview of the organization agents/leaders. |
| | | c. | Leaders and members of the organization. | c. | Workshop with the organization members. |
| Organization profile | Organization profile record | a. | Leaders and members of the organization. | a. | Joint research interview of the organization's agents/leaders. |
| | | b. | Documented registration of the organization. | b. | Workshop with the organization members. |
| Relations among members | Organization members' relation record | a. | Organization members' perceptions. | a. | Workshop with the organization members. |
| | | b. | Enrollment and participation registration of the organization members. | b. | Evidence of the members' enrollment and participation. |

The instrument used to collect information in this categorization was based on a four-point Likert scale with responses "not at all" (0), "somewhat" (1), "a lot" (2), and

"definitely" (3). The scale was designed to avoid neutral answers, identify the positive and negative values of social capital, and mitigate social desirability bias [75].

The tool was adjusted based on expert judgment (the instruments were validated by the following experts: Doctor Francisco Javier Morales, professor at the Polytechnic University of Madrid; Atty. Hernán Rodas Martínez, founder of the Cooperativa Jardín Azuayo of Ecuador; and Atty. Rene Unda Proaño, SDC consultant), the implementation of pilot tests (performed in rural production organizations in Ecuador: La Unión, Nuevo Mundo, Serrag-Ludo, Asosercabo, Cufe, and Ñucanchi Huasi), and the validation of analysis criteria. Validation included the results of local development studies and solidarity finance research that, through factorial analysis, showed the relevance of including socio-organizational variables (appropriate for the social capital concept) in risk evaluation and financial service delivery for a rural population [37].

The application of this proposal for the evaluation of the social capital endowment of rural productive organizations must consider two possible limitations. First, the presence of the researcher and the techniques applied in the process introduce modifications in the dynamics of the relationship of the organization and its members, so it is necessary to reach an analytical interpretation of the meaning of the acts, expressions, behaviors, attitudes, and forms of relationship of the group [76]. Second, the identification of the level of social capital of a rural productive organization does not imply having a comprehensive diagnosis of the group, since the analysis is limited to assessing the quality of social relations in this context.

The proposed method for identifying and evaluating the social capital of a rural production organization was applied to the "The Junta Administradora de Agua Potable y Saneamiento Ambiental Proyecto Nero of Ecuador", and the following factors were considered:

- The importance of the resources managed by the organization;
- Access to safe water and environmental sanitation is directly linked to life, human rights, and community wellness [77]); and
- The contribution of social capital to the organization's sustainability.

In different parts of the world, community organizations that provide water services and environmental sanitation (OCSAS) adopt self-management structures and promote work and collective action that underlie the sustainability of cultural and organizational strength, as evidenced by their social capital endowment [78].

- The representativeness of the case study as a collective action experience:

Until 2016, more than 145,000 community organizations in Latin America participated in the provision of safe water and environmental sanitation services in response to the needs of approximately 70 million people, covering more than 30% of the water requirements in rural and peri-urban areas [79,80]. The Junta Administradora de Agua Potable y Saneamiento Ambiental Proyecto Nero of Ecuador is a community organization that satisfies water requirements in rural and peri-urban zones of Ecuador that are affected by poverty and land desertification.

The case study is relevant because Ecuador, like other countries in the region and around the world, has a rural population that applies community strategies to manage production and satisfy needs. These mechanisms represent capacity and resources that are still not recognized by traditional evaluation and benefit systems and methodologies of formal financial services in rural areas that could facilitate their financial inclusion [29,30].

The reliability of the proposed instrument was estimated for each relationship dimension, component, and variable using Cronbach's alpha, which, for this case study, ranged from 0.77 to 0.83. This range is considered reliable for social research [81]. After adjusting the instrument based on expert observations, pilot tests, evaluation of the relevance of components and variables, and reliability testing, the proposed method was applied to the community organization case study.

The results are presented on the basis of the research objectives. First, the design of the methodology, the study scope, the theoretical focus, and the analytical approximation method used to construct definitions and operationalize the social capital measure are defined.

Second, the fields and dimensions of the research instrument are specified. The incorporation of the social capital evaluation into the financial service delivery methodology is then proposed. Finally, as a case study, the detailed procedure was applied to a community organization, the "Junta Administradora de Agua Potable y Saneamiento Ambiental Proyecto Nero of Ecuador".

## 3. Results and Discussion

### 3.1. Analytical Approximation for Evaluating the Social Capital Endowment of Rural Production Organizations

The analytical approximation of social capital and its measures were applied to a rural production organization case study to illustrate the integration of small family economies that, through collective action, are trying to increase their production scale or manage public goods.

The social capital measure in this field requires the identification of the social position of the group and its members as a historical construction of rural belonging [82], a category from which the territorial relationship context can be analyzed as well as the capacity for promoting the social sustainability and economic structure of the community based on natural resources and environmental services [83].

The collective action effort undertaken by a rural production organization is based on the capacity of the social fabric, and social capital is the resource that allows for the fulfillment of shared objectives for a common benefit. This manifests in relationships within the organizational structure and in the contextual connections formed both inside and outside the operational territory [84].

The process used to identify social capital components must account for the territorial context, the organization profile, and the relationships among members that, in terms of benefits or drawbacks, can affect the collective action of the organized group. The construct designed for this purpose can be viewed with a socioeconomic theoretical focus (Figure 1). A scientific approach that explains the true complexity of social phenomena should incorporate sociological, economic, cultural, and political elements in order for the capacity of the social fabric in rural communities to be understood [67].

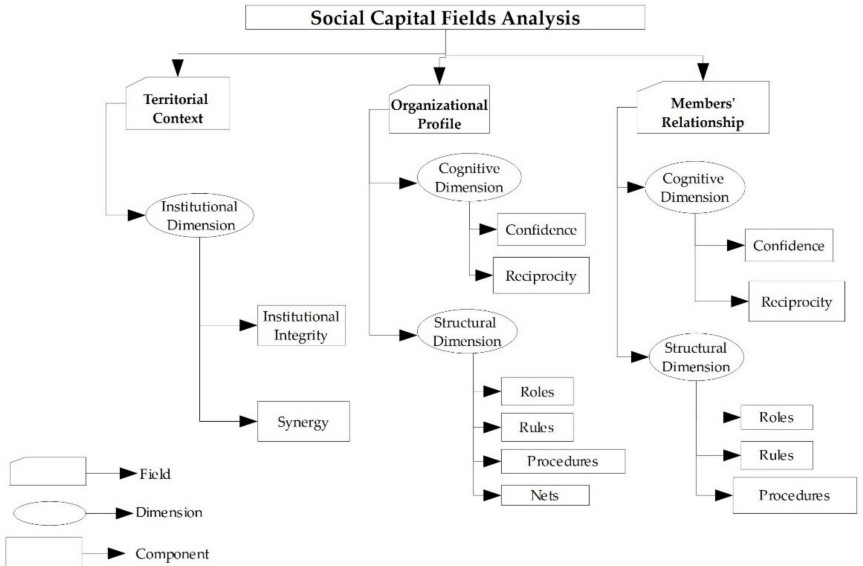

**Figure 1.** Social capital fields, dimensions, and components.

The established considerations are suitable for rural production organizations in different analysis contexts within the study field because they describe and explain, in a general and systemic way, the content of the concept. They also integrate aspects of the territorial context, the organization profile, and the relationship among members, combining exogenous elements of the macro field with endogenous aspects of the internal structure. The approach captures the multidimensional context by including institutional [10,85–88], cognitive [89], and structural [90–92] variables.

In the territorial field, the treatment of the institutional dimension of social capital collects second generation neo-institutional approaches. Coleman [91], Evans [86], Woolcock [93] and Putnam [89] emphasize the importance of the quality of institutions to value social capital and promote local development because of the positive effects over the uncertainty treatment, the risk, the opportunistic behavior control, and the reduction in transaction costs.

In line with these considerations, the institutional dimension of the social capital of a rural production organization includes territorial contextual factors that affect the local operation. This dimension also includes the capacity of the state to provide goods and basic services while creating synergistic relationships with the organization to promote the local development of public policies.

The organization profile and the members' relationship fields contribute to the analysis of the structural and cognitive dimensions of social capital. The organization profile considers the group as a social or individual actor, as described by Burt [46], and identifies the group capacity for developing strong relationship ties among members and weak links with other actors [87]. In addition, the analysis of sociocentric relationships evaluates the union and prevalence of social capital in horizontal relationships among members [92].

The cognitive dimension incorporates confidence and reciprocity components in the fields of the organization profile and member relationships. In the organization profile field, the cognitive dimension evaluates the institutional capacity for building a group identity and creating reciprocity practices. The members' relationship field identifies interpersonal levels of trust and reciprocity [94].

In accordance with the concepts of Uphoff [95], Zimmermann [96], and Burt [68], the structural dimension is addressed in the organization profile and members' relationship fields. Within the organization profile, the structural aspect evaluates the quality of the management of the organization in assigning roles, defining rules, establishing management procedures, and participating in networks.

In the members' relationship field, the structural components identify the approval level, the members' recognition and negotiation of roles, and the rules and management procedures established by the organization.

After the field and dimensions are defined, the measurement of social capital requires the selection of variables and indicators that describe and explain the strengths of the organizational social fabric related to the capacity for creating collective action.

Therefore, the integral components of the institutional dimension include the state's capacity to provide goods and basic services, communication infrastructure with adequate coverage and quality, fixed and mobile telephony, Internet service, safe drinking and irrigation water, education, health, environmental sanitation, and security infrastructure as well as a level of cultural relevance in the delivery of education and public health services. The synergic component includes variables that represent active citizenship, which reflect the level of civic engagement of the organization with its community through participation in the local development of public policies (Figure 2).

The territorial context record is the instrument used to collect information about the proposed categories for evaluating the institutional dimension of social capital in rural production organizations.

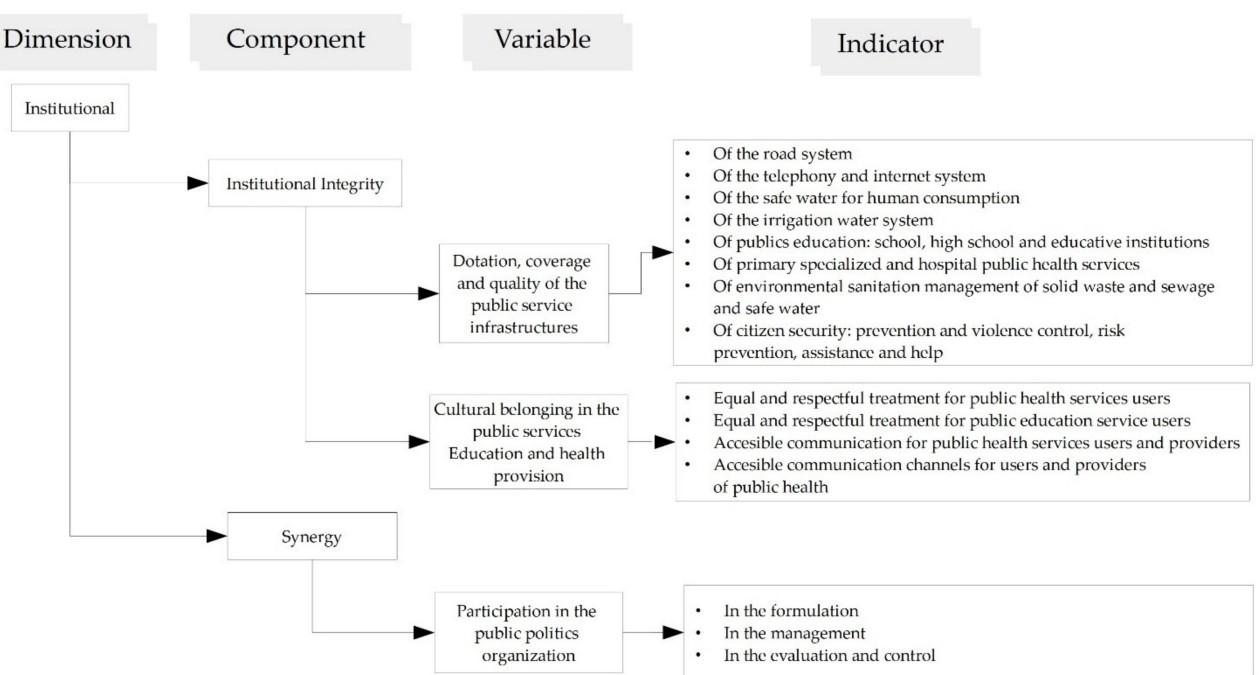

**Figure 2.** Institutional dimension of social capital of the territorial operating context.

According to the cognitive dimension (Figures 3 and 4), the variables and indicators represent the confidence and reciprocity components and identify the types of relationships that prevail within the organization as the cultural expression of social and ethical values in the organization profile and members' relationship fields. The application of common variables between the organization profile and members' relationship fields allows for the comparison of perspectives and the correlation between the institutional focus indicated in the organization profile and the members' perceptions of the organization. The variables of the confidence component include group consolidation, collective identity construction, capacity for creating collective learning, and the authority of leadership. The reciprocity component incorporates variables that promote mutual support and define mechanisms for punishing opportunistic behavior.

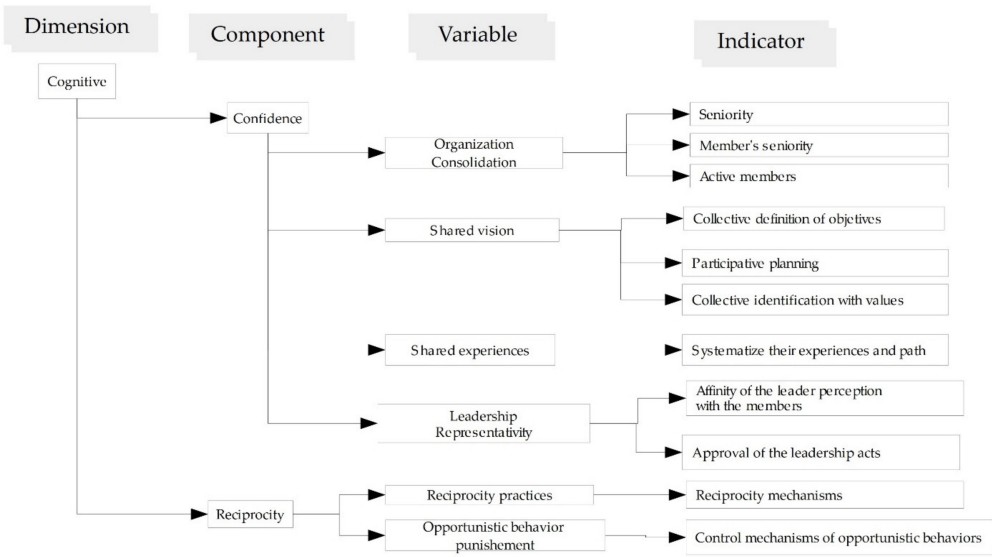

**Figure 3.** Cognitive dimension of social capital in the organization profile field.

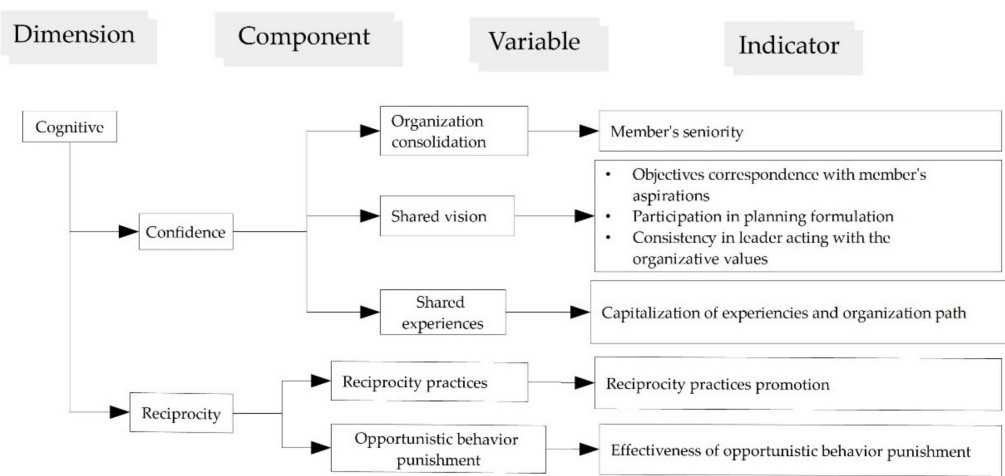

**Figure 4.** Cognitive dimension of social capital in the organization members' relationship field.

The variable and indicator definitions of the structural dimension of social capital in the organization profile and members' relationship fields (Figures 5 and 6) can be used to assess whether the internal governance structure is arranged with clear, useful, and effective norms and rules that promote the democratic participation of the members in resource management, decision making, internal communication processes, and conflict resolution. At the organization profile level, the structural dimension identifies the participation level and the consensus for defining roles, rules/norms, resource management procedures, and decision making as well as the internal communication quality, the conflict resolution capacity, and the organization's relationship with other actors related to the effective size, the density, and the heterogeneity of the relationships.

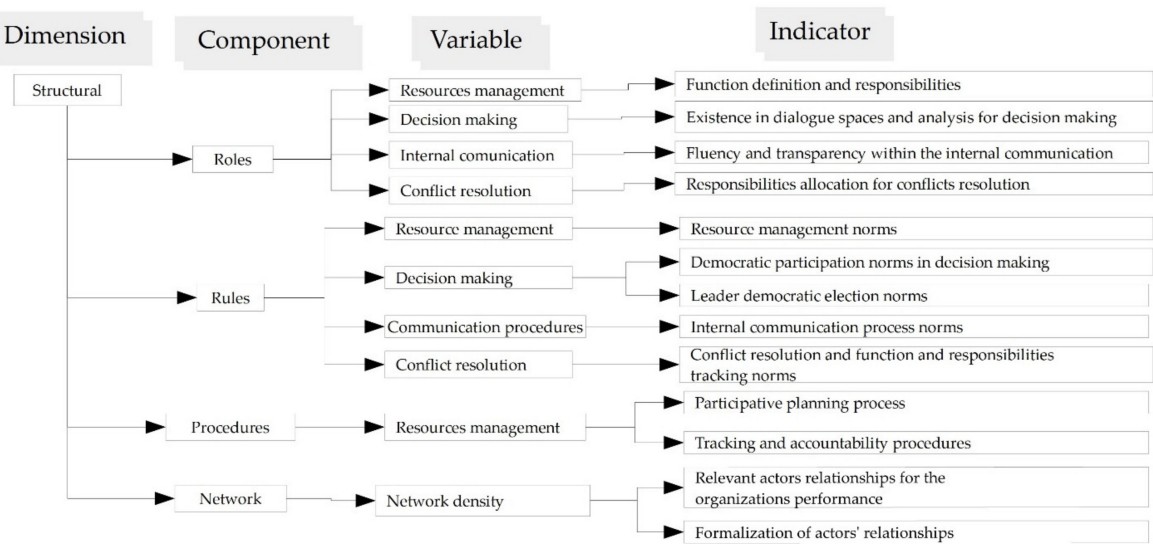

**Figure 5.** Structural dimension of social capital of the organization profile field.

In the relationships among members, the structural dimension identifies the role of rules/norms, the level of approval, and member satisfaction with the procedures for resource management, decision making, internal communication, and conflict resolution.

The organization profile and members' relationship measurements were designed to collect information to support the evaluation of the cognitive and structural dimensions of the social capital of rural production organizations.

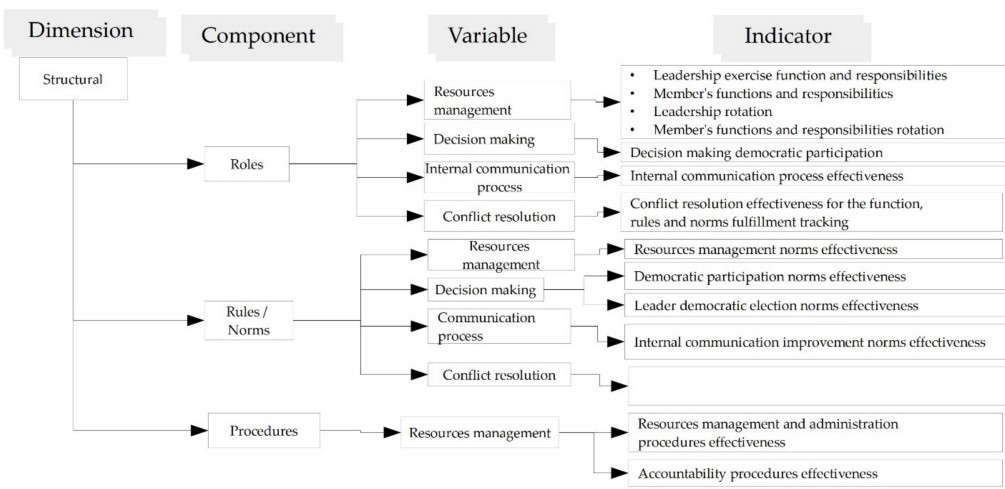

**Figure 6.** Structural dimension of social capital of the organization relationships among members.

*3.2. Results of Integration and the Identification of Social Capital Level in Rural Production Organizations*

Once the research techniques have been applied according to the analytical approach established by the construct, the integration process of the results begins. The proposal applied a weighted relational integration procedure as follows:

The indicators represent the greater degree of disaggregation of the information, and their value oscillates according to a 4-point Likert scale (0; 1; 2; 3) with 3 being the highest possible value. To obtain the value of each variable, the scores of the indicators that comprise it are added and the result is relativized according to the possible maximum. This result represents the positive social capital ($n$), and its reciprocal ($1 - n$) identifies the corresponding negative value. The difference between the positive and negative values of social capital $[n - (1 - n)]$ represents the social capital endowment (d), which is defined as a high, medium, or low level in the positive $\{0 < [n - (1 - n)] \leq 1\}$ or negative $\{-1 \leq [n - (1 - n)] < 0\}$ valuation range. The valuation range includes the possibility of positive, negative, or no social capital endowment (Table 2).

**Table 2.** Social capital endowment range.

| $d = [n - (1 - n)]$ | | | | | | |
|---|---|---|---|---|---|---|
| **Negative Range** | | | **Absence of Relation** | **Positive Range** | | |
| **High** | **Medium** | **Low** | | **Low** | **Medium** | **High** |
| **($-1 \leq$ d $< -0.667$)** | ($-0.667 \leq$ d $< -0.333$) | ($-0.333 \leq$ d $< 0$) | 0 | ($0 <$ d $\leq 0.333$) | ($0.333 <$ d $\leq 0.667$) | ($0.667 <$ d $\leq 1$) |

Each component is made up of one or more variables, so that, to obtain its value, the score of such variables is averaged. In the same way, the dimensions are made up of one or more components, and their scores express the average valuation of such components. The first dimension to be calculated is the territorial dimension. The result is placed in the positive or negative range of high, medium, or low level of social capital endowment. This value defines the weightings to be applied to assess the cognitive and structural components of the organization's profile dimension.

Then, the value of the organization's profile, located on the positive or negative range of the social capital endowment, defines the weightings to be applied to assess the cognitive and structural components of the 'relationship between members' dimension.

Finally, to obtain the global organization´s social capital endowment, the results of the dimensions 'profile of the organization' and 'relationship between members' were averaged.

The social capital value fields include the organization's exogenous and endogenous variables, which have a logical relationship and can be expressed as an algorithm that

integrates the results by field. In this way, the territorial context contributes exogenous variables to an institutional dimension that affects the organization's productivity and facilitates development possibilities in the community, which determines the work environment quality as defined by Krugman [97]. The organization profile and members' relationship fields indicate the endogenous cognitive and structural dimensions of social capital.

The fields represent values attributable to interrelated spatial contexts: the territorial context affects the organization profile, and, simultaneously, the local actors, the organization, and its members modify the institutional components of the territorial context. On the other hand, the organization profile affects the quality of relationships among members, and this relationship contributes to the definition of the organization profile (Figure 7).

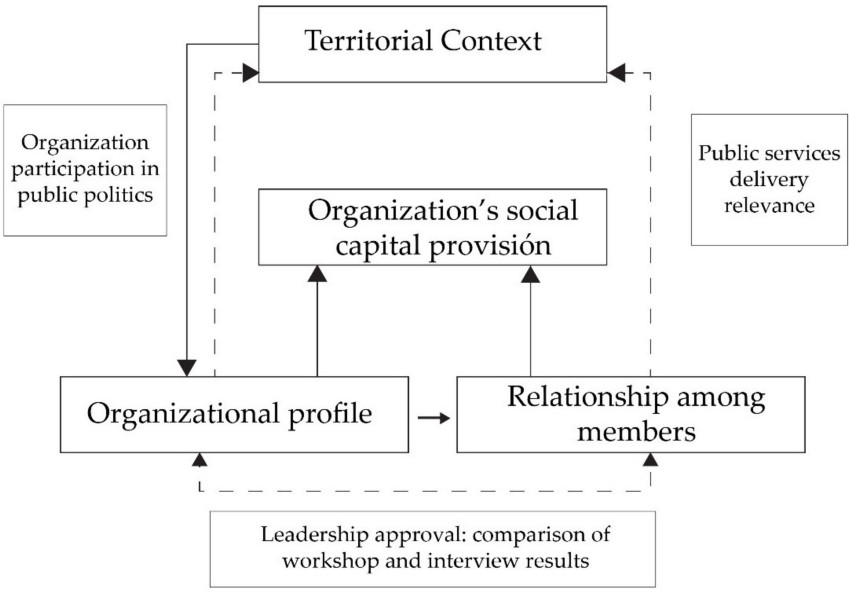

**Figure 7.** Field interrelation for social capital evaluation.

The proposed approach integrates the results starting with the territorial context field of action, the value of which determines the considerations applied to the cognitive and structural dimensions in the organization profile field. At the same time, the social capital of the organization profile affects the considerations applied in evaluating the structural and cognitive dimensions in the member relationship field. In this way, the social capital endowment is calculated in the organization profile and member relationship fields. These values are averaged, and the result represents the social capital endowment of the organization (Table 3).

*3.3. Proposed Method for Incorporating the Social Capital of a Rural Production Organization into the Financial Service Delivery Methodology and Risk Qualification*

If social capital is a production resource that is characterized by relationships and creates the capacity for collective action, then its incorporation as a variable in financial analysis to improve the risks, costs, and technological management of rural population initiatives is meaningful. The inclusion of social capital in the criteria that define the risk profiles of small rural producers will enable the application of group evaluations and the coordination of timing and movement. It will also reduce unit costs and facilitate the implementation of processes and systems that take advantage of the proximity that community organization local networks provide.

The social capital endowment of an organization can be classified as high, medium, or low within the positive or negative range of values, so it can be included in the financial risk qualification process applied by financial institutions, according to the recommendations provided by the Basel Committee of Banking Supervision Therefore, it is possible to

incorporate the social capital of a rural production organization into financial service delivery methodologies and risk qualification.

**Table 3.** Interpretation of results and the identification of the social capital endowment of the organization.

| | Level of Social Capital Provision | Dimension Balancing for the Organization Profile | | Level of Social Capital Provision | Dimension Balancing for the Relationships Among Members | |
|---|---|---|---|---|---|---|
| | Territorial context | Cognitive | Structural | Organization profile | Cognitive | Structural |
| Social capital territorial context | High positive ( $0.667 < d \leq 1$ ) | 25% | 75% | High positive ($0.667 < d \leq 1$) | 25% | 75% |
| | Medium positive ($0.333 < d \leq 0.667$) | 35% | 65% | Medium positive ($0.333 < d \leq 0.667$) | 35% | 65% |
| | Low positive ($0 < d \leq 0.333$) | 45% | 55% | Low positive ($0 < d \leq 0.333$) | 45% | 55% |
| | Absence 0 | There is no interaction | | Absence 0 | There is no interaction | |
| | Low negative ($-0.333 \leq d < 0$) | 55% | 45% | Low negative ($-0.333 \leq d < 0$) | 55% | 45% |
| | Medium negative ($-0.667 \leq d < -0.333$) | 65% | 35% | Medium negative ($-0.667 \leq d < -0.333$) | 65% | 35% |
| | High negative ($-1 \leq d < -0.667$) | 75% | 25% | High negative ($-1 \leq d < -0.667$) | 75% | 25% |
| | Organization profile's social capital | | | Relationship among members' social capital | | |
| | Social capital average provision of the organization profile and the relationship among members | | | | | |
| | Social capital provision for the organization | | | | | |

The proposed method relates social capital to risk qualification for an individual profile. A positive value of social capital endowment improves the risk profile, and a negative value worsens it. The intensity of the effect depends on the social capital level; thus, if the endowment value is low (positive or negative), then the profile will depend on the risk value qualification. A medium positive value will result in a low level of perceived risk in the qualification, and a medium negative value will increase the level of perceived risk. A high positive endowment value improves the risk perception by two levels, and a high negative value decreases it by two levels (Table 4).

Incorporating the social capital of an organization into a financial service methodology means recognizing the importance of evaluating individual economic and relational sociological variables. This expands the approximation and definition of the risk profile of small rural producers and favors the construction of inclusive financial ecosystems.

*3.4. Case Study Application: The Junta Administradora de Agua Potable y Saneamiento Ambiental Proyecto Nero of Ecuador*

The proposed method of social capital evaluation was applied to a community organization: "Junta Administradora de Agua Potable y Saneamiento Ambiental Proyecto Nero of Ecuador". In this country, community organizations in associative, cooperative, and single-family subsistence entrepreneurship sectors are classified as popular and solidarity economies (Law of the Popular and Solidarity Economy, Ecuador). Their registry, regulation, and control are the responsibility of the Popular and Solidarity Economy Superintendence, which, in January 2021, reported the existence of 15,245 nonfinancial organizations of this type, 68.3% of which were in rural zones wherein more than the 50% of the population lives in poverty [98].

The delivery of safe water and environmental sanitation services for rural and peri-urban sectors in Ecuador is significantly fragmented because it is a small country with a rural population that accounts for less than 32% of the total population [99]. There are more than 7000 community organizations, with 2803 Juntas that provide services to approximately 3.5 million users [100].

**Table 4.** Effect of social capital on risk profiles for financial service delivery.

| | | Social Capital Provision Level | | Risk Qualification | Risk Profile |
|---|---|---|---|---|---|
| $d = [n - (1 - n)]$ | Positive range | High | $(0.667 < d \leq 1)$ | High | Low |
| | | | | Medium | Low |
| | | | | Low | Low |
| | | Medium | $(0.333 < d \leq 0.667)$ | High | Medium |
| | | | | Medium | Low |
| | | | | Low | Low |
| | | Low | $(0 < d \leq 0.333)$ | High | High |
| | | | | Medium | Medium |
| | | | | Low | Low |
| | | Absence of relation | 0 | | |
| | Negative range | High | $(-1 \leq d < -0.667)$ | Low | Low |
| | | | | Medium | Medium |
| | | | | High | High |
| | | Medium | $(-0.667 \leq d < -0.333)$ | Low | Low |
| | | | | Medium | High |
| | | | | High | High |
| | | Low | $(-0.333 \leq d < 0)$ | Low | High |
| | | | | Medium | High |
| | | | | High | High |

The selected board is a collective action initiative formed by the Farmer Organization of Turi (OCT), which has operated since 1982. In 1985, the Autonomous Community System Nero Project was established to collaboratively meet the need for safe water in a rural zone affected by deforestation and land desertification. With the establishment of the Organic Law of Water Resources, Uses and Utilization in 2014, its legal identity was transferred to the Junta by integrating fortythree communities, 7.4 thousand members, and 30 thousand direct consumers in rural sectors such as Turi, Baños, and el Valle (the parish is the smallest political administrative category in Ecuador) of "Cuenca" canton, Azuay province.

Although the socio-organizational structure of the board and other community organizations of water and environmental sanitation (OCSAS) presents management aspects, all of them are controlled by the leadership under a self-management model, and the collective work of the users reflects the collective action capacity of the group to exercise this right [78].

The governmental and administrative structure of the Junta is set up as a network, laying the social foundation of the organization. Thus, users of the system can democratically exercise their right to participate in and control the organization and the actions of its leaders and administrators. Three governmental and administrative levels are defined: general assembly, directory, and local committee.

The Consumers General Assembly is the higher level of constitutive direction and decision making and constitutes a setting for social representation by the presidents of local committees and the members of the directory. The directory encompasses the administration management, financial, commercial, operational, and technical bodies; it is managed by a president, a vice president, a secretary, a treasurer, and six main spokesmen with their respective alternates, representing the leadership space that the members influence by voting. Each community has a local committee, which has the same structure as the

directory and is managed by members of the community through democratic elections, enabling the oversight and coordination of local politics, programs, and project execution.

After the Junta operational field and the government structure were determined, the proposed research techniques were applied to measure the social capital in each analysis field. The database is presented in the supplementary material.

The territorial context affects the institutional dimension of social capital. Therefore, the provision, coverage, and quality of public service infrastructure were identified and recorded in the Integral Diagnosis and Local Government Territorial Development and Regulation Plan. The organization's leaders and members were consulted with regard to the cultural relevance level in the public service delivery and group work capacity of the Junta with the state and the local government.

In order to identify the social capital within the profile field of the organization, opinions and evidence were collected from group leaders. To evaluate the social capital endowment of relationships among members in the field, the local committee members' perceptions about the quality of internal relationships were collected. The levels of representativeness and approval of management by the board leaders were identified, and the results of the group interview with the leaders were compared with those collected in the workshop with the organization members.

The result of valuing the institutional social capital in the territorial acting context of the Junta was negative, and there were limited conditions for the local development promotion in both the integrity components and synergy related to the deficient perception of cultural belonging in the public services delivery of education and health as well as the weakness of the Junta participation in the promotion of public politics of local development (Table 5).

**Table 5.** Institutional dimension of social capital within the territorial context of the Junta's operations.

| Analysis Categories | | | | Variable Score | Positive Social Capital ($n$) | Negative Social Capital ($1 - n$) | Social Capital Provision | | | |
|---|---|---|---|---|---|---|---|---|---|---|
| Field | Dimension | Component | Variable | | | | Variable $[n - (1 - n)]$ | Component | Dimension | Field |
| Territorial | | | | | | | | | | −0.03 |
| | Institutional | | | | | | | | −0.03 | |
| | | Integrity | | | | | | −0.39 | | |
| | | | Provision, coverage and quality of public service infrastructures. | 16/30 | 0.53 | 0.47 | 0.06 | | | |
| | | | Cultural belonging in education and health public services provision. | 1/12 | 0.08 | 0.92 | −0.84 | | | |
| | | Synergy | | | | | | 0.34 | | |
| | | | Junta's participation in the public politics of local development promotion. | 6/9 | 0.67 | 0.33 | 0.34 | | | |

The values of the cognitive and structural dimensions of social capital in the Junta's organization profile were positive, and the cognitive dimension was slightly higher. The weakness of the links reduced the structural dimension value and caused isolation of the organization; therefore, the cognitive dimension, the shared vision, and the leadership representation maintained the internal cohesion of the organization (Table 6).

The value attributed to relationships among members was in the high positive range in the cognitive dimension of social capital: the group's identity and cohesion were reinforced by the effective participation of the members, the clear definition of objectives, and the common vision. However, the structural dimension revealed limitations in the definition of rules/norms, the effectiveness of conflict resolution procedures, and the punishment of opportunistic behaviors (Table 7).

**Table 6.** Cognitive and structural dimensions of the social capital of the board's organization profile.

| Analysis Categories | | | | Variable Score | Positive Social Capital (*n*) | Negative Social Capital (1 − *n*) | Social Capital Provision | | |
|---|---|---|---|---|---|---|---|---|---|
| Field | Dimension | Component | Variable | | | | Variable [*n* − (1 − *n*)] | Component | Dimension |
| **Organization profile** | Cognitive | | | | | | | | 0.467 |
| | | Confidence | | | | | | 0.600 | |
| | | | Junta's consolidation | 4/6 | 0.667 | 0.333 | 0.333 | | |
| | | | Shared experiences | 2/3 | 0.667 | 0.333 | 0.333 | | |
| | | | Leadership representativity | 13/15 | 0.867 | 0.133 | 0.733 | | |
| | | | Shared vision | 9/9 | 1.000 | - | 1.000 | | |
| | | Reciprocity | | | | | | 0.333 | |
| | | | Reciprocity practices | 2/3 | 0.667 | 0.333 | 0.333 | | |
| | | | Opportunistic behavior punishment | 2/3 | 0.667 | 0.333 | 0.333 | | |
| | Structural | | | | | | | | 0.333 |
| | | Roles | | | | | | 0.667 | |
| | | | Resources management | 2/3 | 0.667 | 0.333 | 0.333 | | |
| | | | Internal communication process | 3/3 | 1.000 | - | 1.000 | | |
| | | | Conflict resolution | 2/3 | 0.667 | 0.333 | 0.333 | | |
| | | | Decision making | 3/3 | 1.000 | - | 1.000 | | |
| | | Rules/norms | | | | | | 0.667 | |
| | | | Resources management | 3/3 | 1.000 | - | 1.000 | | |
| | | | Internal communication process | 2/3 | 0.667 | 0.333 | 0.333 | | |
| | | | Conflict resolution | 2/3 | 0.667 | 0.333 | 0.333 | | |
| | | | Decision making | 6/6 | 1.000 | - | 1.000 | | |
| | | Procedures | | | | | | 1.000 | |
| | | | Resources management | 6/6 | 1.000 | - | 1.000 | | |
| | | Nets | | | | | | −1.000 | |
| | | | Net density | 0/6 | - | 1.000 | −1.000 | | |

**Table 7.** Cognitive and structural dimensions of social capital within the relationships among board members.

| Analysis Categories | | | | Variable Score | Positive Social Capital (*n*) | Negative Social Capital (1 − *n*) | Social Capital Provision | | |
|---|---|---|---|---|---|---|---|---|---|
| Field | Dimension | Component | Variable | | | | Variable [*n* − (1 − *n*)] | Component | Dimension |
| Relationships among members | Cognitive | | | | | | | | 0.620 |
| | | Confidence | | | | | | 0.301 | |
| | | | Junta's consolidation | 8/33 | 0.242 | 0.757 | −0.515 | | |
| | | | Shared experiences | 26/33 | 0.787 | 0.212 | 0.575 | | |
| | | | Shared vision | 152/165 | 0.921 | 0.078 | 0.842 | | |
| | | Reciprocity | | | | | | 0.939 | |
| | | | Reciprocity practices | 31/33 | 0.939 | 0.060 | 0.878 | | |
| | | | Opportunistic behavior punishments | 33/33 | 1.000 | - | 1.000 | | |
| | Structural | | | | | | | | 0.515 |
| | | Roles | | | | | | 0.502 | |
| | | | Resources management | 148/231 | 0.640 | 0.359 | 0.281 | | |
| | | | Internal communication process | 27/33 | 0.818 | 0.181 | 0.636 | | |
| | | | Conflict resolution | 19/33 | 0.575 | 0.424 | 0.151 | | |
| | | | Decision making | 32/33 | 0.969 | 0.030 | 0.939 | | |
| | | Rules/norms | | | | | | 0.227 | |
| | | | Resources management | 26/33 | 0.787 | 0.212 | 0.575 | | |
| | | | Internal communication process | 20/33 | 0.606 | 0.393 | 0.212 | | |
| | | | Conflict resolution | 12/33 | 0.363 | 0.636 | −0.272 | | |
| | | | Decision making | 46/66 | 0.696 | 0.303 | 0.393 | | |
| | | Procedures | | | | | | 0.818 | |
| | | | Resources management | 30/33 | 0.909 | 0.090 | 0.818 | | |

The board's institutional social capital was evaluated in the low negative range. The applied values at this level were 0.45 and 0.55 for the cognitive and structural dimensions, respectively, which placed the social capital profile of the board in the medium positive range. For the relationships among members, the social capital was in the medium positive range. Finally, the general provision level of the board reached a value of 0.48, corresponding to the medium positive level (Figure 8).

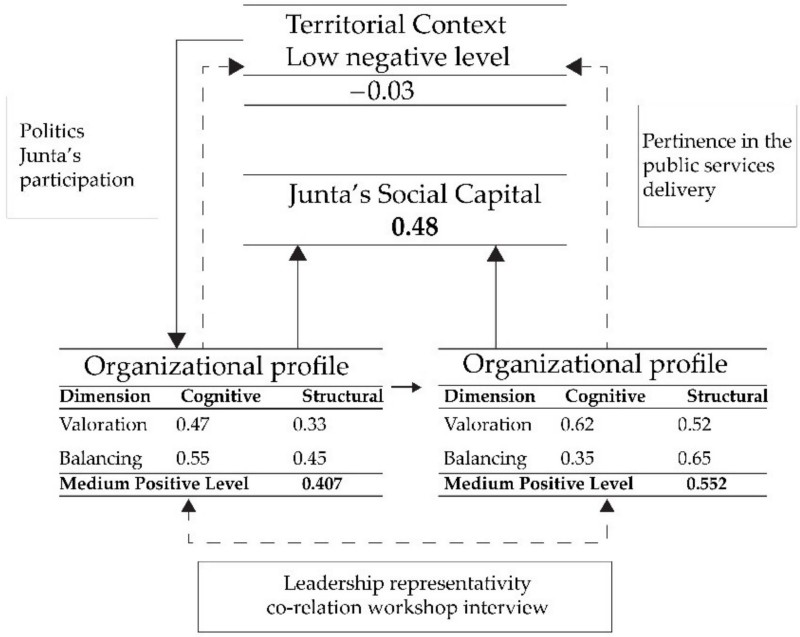

**Figure 8.** Integration of results and identification of the board's social capital endowment level.

A member of the board with a medium risk qualification will have a profile of low risk, while a member with high risk qualification will have a medium risk profile, according to the effect of social capital levels on risk qualification and the profile definition presented in Table 4.

## 4. Final Considerations

The research experience generates reflections on the contribution of social capital to create inclusive financial ecosystems of sustainable local development and to strengthen the social function of solidarity finance institutions in rural areas. In these spaces, social capital raises the capacity for collective action, strengthens processes of social cohesion, and creates integration links that give rise to situations of synergy between actors. Thus, for small rural producers, social capital represents part of their intangible heritage, which favors the access and responsible use of resources, provides support, and promotes social learning.

Social capital, by being constituted by relational categories, strengthens the capacity for analysis and understanding of the facts, being able to contribute to the effectiveness of initiatives focused on improving living conditions in different contexts. Measurement efforts will make it possible to identify the quality of social relations and the strengths and weaknesses of the social fabric, thus promoting strategic thinking and the capacity for political action.

In this way, it was identified that the social capital valuation procedures require starting with an approach to the ontological components that support the social measurement space, which in the case of this proposal was community production in common property organizations. Once the construct that synthesizes the dimensions, components, and relevant variables that express the social capital is defined, and maintaining coherence with the conceptual approach used, the objective of measuring such a construct is achieved.

The selection of measurement methods, techniques, and tools must be adapted to the context conditions as they are not neutral and could affect the measurement results. Likewise, the researcher must assume an interpretive analytical position of the different expressions of social capital in the specific context, since it is a non-statistical study of generalization, temporally and historically defined.

The existence of social capital makes sense in a defined social space, so its assessment must relate to the field of measurement and development of the actor considered. Making its measurement feasible means addressing the complexity of the concept in terms of the complementarity and interrelation between different levels and dimensions of social relationship as well as including the possible existence of positive or negative relationship scenarios.

For community organizations, measuring their social capital endowment means an effort of introspective reflection that makes the quality of the group's social relationship visible. For solidarity finance institutions, identifying the level of social capital of rural productive organizations opens the possibility of recognizing and incorporating the value of non-monetizable relational categories in their service provision procedure. Furthermore, for the investigative task, the proposal contributes to the development of methodologies for approaching complex concepts.

**Supplementary Materials:** The following are available online at https://www.mdpi.com/article/10.3390/su13137067/s1.

**Author Contributions:** Conceptualization, J.S.; Data curation, J.S.; Formal analysis, J.S.; Investigation, J.S.; Methodology, J.S.; Project administration, J.S.; Resources, J.S.; Supervision, S.S.-M.; Validation, J.S.; Visualization, J.S.; Writing—original draft, J.S.; Writing—review & editing J.S. and S.S.-M. All authors have read and agreed to the published version of the manuscript.

**Funding:** This research received no external funding.

**Institutional Review Board Statement:** Not applicable.

**Informed Consent Statement:** Not applicable.

**Data Availability Statement:** Not applicable.

**Conflicts of Interest:** The authors declare no conflict of interest.

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
