# Peer review of "Social Capital as an Inclusion Tool from a Solidarity Finance Angle"

_sustainability, doi:10.3390/su13137067_

Round 1
Reviewer 1 Report
The authors propose a new framework to promote financial service delivery of solidarity finance institutions through the evaluation of social capital in rural production organizations.
The literature review is well constructed and presents relevant developments in the field of social capital, social finance and related topics started from the first years to the latest studies.
The research methodology is well presented and suited for the studied topic. The conclusions are supported by the results.
Author Response
Thank you for the time and work dedicated to reviewing our work.
Reviewer 2 Report
This is an interesting and well-written article. The authors provide a thorough literature review, followed by details on method, method development, results and discussion, including visually easy to follow graphics. The discussion of social capital and its importance for poor rural communities is interesting and very useful. The importance of solidarity finance to a rural community is convincingly described, and its role and importance in generating inclusion and value is clearly laid out.
Overall, this is a useful article, an important piece of research that contribute to the discussion of the importance of social capital for rural communities, valuation of social capital, and the development of indicators for measurement. The model/method presented with the various flow charts provides a good overview.
There are some minor considerations however.
The article simply refers to rural environments. It is unclear to the reader from the beginning and throughout whether the research only applies to developing, poor, low-income environments. Consistently throughout, the article refers to rural populations as being poor. There are of course many parts of the world where rural populations cannot be categorized as poor, or where poverty is not widespread, but where a system that incorporates consideration of social capital is still useful. There seem to be an assumption made that rural communities are poor; and therefore that the focus is on developing countries. It would be useful if this is spelled out or clarified. Also, a note on the usefulness of this system/methodology for rural communities/remote/small communities in general.
Having said that, the method developed for the valuation of social capital of rural production organizations is useful. However, as for the indicators - they are very broadly defined, and there are several identified. It is unclear whether these indicators are valid for all contexts, i.e. rural settings beyond developing, in transition, or low income countries; and also how selection between indicators would be determined - what are the selection criteria used? Also, it is unclear what the details are in terms of the process of co-designing these categories and indicators with the relevant stakeholders. The approach for measuring social capital could be further developed by devising more specific measures of the indicators and presenting data for this measurement. Perhaps a brief note on selection, data, and pitfalls with the method might be useful.
While the graphics and data tables are useful, and clearly presented within the graphical illustration, there still seem to be some detail missing in the text on how these were arrived at.
About risk evaluation, might be useful to add a brief sentence on how you define risk.
There is a typing error in Table 4 (hight).
How the results are arrived (e.g. table 5) could be better described, further spelled out, and elaborated on.
Section 4 on final considerations: this section is presented in bullet format which seems less appealing and creates a different flow than for the rest of the article. Suggest to write it out and remove the bullets, although this is probably just a matter of taste.
Author Response
Dear reviewer,
Thank you for the time and work dedicated to reviewing our work.
Observing the comments to the best of our knowledge, we have addressed all of the suggestions and included the necessary modifications (in light blue) in the revised version of our article.
Please see the attachment, for finding a detailed list of modifications.

Reviewer 3 Report
The paper presents an analytical tool to include the cognitive elements of social capital in what is called "solidarity finance". The empirical analysis is promising but is a bit muddled up in the way it is presented and it is not at the moment grounded in the relevant literature.
Areas for improvement includes:
- the theoretical framework needs to be totally re-written in light of the English literature around microfinance. "Solidarity finance" is not well-defined and the microfinance literature needs to be included as a benchmark of analysis.
- the theoretical framework refers to Ostrom's work which should sit as the main theoretical background with her key principles of co-production and management of the the commons used as criteria of evaluation of the empirical analysis - it is not clear how the analysis sits within Ostrom's work, or how it departs from it.
- the writing of the paper is very heavy with long sentences and an exposition of ideas one after the other without a consistent argumentation throughout the theoretical part - I suggest to be supported by a native English writer to be able to unpack your arguments.
- the issue of power is missing from the relational framework, yet its impact on the cognitive elements are crucial - the hierarchical structure of the organisation also call for the issue of power to be addressed
Author Response

(The authors gave the same response as above.)

Round 2
Reviewer 3 Report
None